# Microalgae from Biorefinery as Potential Protein Source for Siberian Sturgeon (*A. baerii*) Aquafeed

**Tiziana Bongiorno [1], Luciano Foglio [1], Lorenzo Proietti [1], Mauro Vasconi [2],**
**Annalaura Lopez [2], Andrea Pizzera [3], Domenico Carminati [4], Aldo Tava [4],**
**Antonio Jesús Vizcaíno [5], Francisco Javier Alarcón [5], Elena Ficara [3] and Katia Parati [1,\***

1    Istituto Sperimentale Italiano L. Spallanzani—Loc. La Quercia, 26027 Rivolta d'Adda (CR), Italy; t.bongiorno@tstechnologies.it (T.B.); luciano.foglio@istitutospallanzani.it (L.F.); lorenzo.proietti@istitutospallanzani.it (L.P.)

2    Department of Veterinary Medicine, University of Milan, 26900 Lodi, Italy; mauro.vasconi@unimi.it (M.V.); annalaura.lopez@unimi.it (A.L.)

3    Department of Civil and Environmental Engineering, Polytechnic University of Milan, 20133 Milano, Italy; andrea.pizzera88@gmail.com (A.P.); elena.ficara@polimi.it (E.F.)

4    Council for Agricultural Research and Economics, Research Centre for Animal Production and Aquaculture (CREA-ZA), 26900 Lodi, Italy; domenico.carminati@crea.gov.it (D.C.); aldo.tava@crea.gov.it (A.T.)

5    Department of Biology and Geology, Ceimar-University of Almería, 04120 Almería, Spain; avt552@ual.es (A.J.V.); falarcon@ual.es (F.J.A.)

\*    Correspondence: katia.parati@istitutospallanzani.it; Tel.: +39-0363-78883

**Abstract:** The demand for aquafeed is expected to increase in the coming years and new ingredients will be needed to compensate for the low fish meal and oil availability. Microalgae represent a promising matrix for the future aquafeed formulation, however, the high production cost hinders its application. The use of microalgae from biorefinery would reduce the disposal costs for microalgae production. The present study aimed to (i) verify the growth of microalgae on digestate coming from pig farming and (ii) evaluate their potential valorization as dietary ingredient in aquafeed according to a Circular Bioeconomy approach. For these purposes, a microalgae biomass was produced on an outdoor raceway reactor supplied with digestate and used for partial replacement (10% of the diet) in aquafeed for Siberian sturgeon fingerlings (*Acipenser baerii*). The results obtained confirm the feasibility for growing microalgae on digestate with satisfactory productivity (6.2 gDM m$^{-2}$ d$^{-1}$), nutrient removal efficiency and Chemical Oxygen Demand reduction; moreover, the feeding trial carried out showed similar results between experimental and control groups ($p > 0.05$), in term of growth performance, somatic indices, fillet nutritional composition and intestinal functionality, to indicate that microalgae from biorefinery could be used as protein source in Siberian sturgeon aquafeed.

**Keywords:** sturgeon; aquafeed; biorefinery; digestate; microalgae

---

## 1. Introduction

According to data reported by FAO [1] aquaculture supplies about half of the fish intended for human consumption and the trend is increasing significantly in the coming decades.

These data are in contrast with the significant reduction of fish from fisheries, due to an excessive depletion of wild fish stocks. Consequently, the demand for aquafeed has grown exponentially in the last decade and a further increase is expected in coming years [2]. Fish meal (FM) and fish oil (FO) are the raw ingredients commonly used to prepare feed for carnivorous fish species, due to their excellent nutritional properties such as high-quality protein content, long chain omega-3 polyunsaturated fatty

acids (LC-PUFA) and high digestibility and palatability [3]. However, there are several limits to their future use, since high quality FM and FO derive from wild fish of which there is a growing demand even from emerging economies. Thus, the availability and consequently the price of these ingredients is unstable and growing substantially [4].

Therefore, the environmental, ethical and economic unsustainability, makes the future use of fish meal and oil in the aquafeed uncertain [3,5]. In order to face this problem, in the last decades the interest of researchers and aquafeed companies has focused on testing the feasibility of including into the feed novel ingredients of vegetable origin as a partial replacement of the protein and lipid sources of marine animal origin [5]. However, these ingredients have various nutritional disadvantages, including an unbalanced amino acid composition and the presence of anti-nutritional factors that can result in poor digestibility and palatability of the feed. Moreover, the vegetable resources demanded by the food and feed market is expected to rise, thus limiting their availability for aquaculture. Among the potential novel ingredients in aquaculture, microalgal biomasses is a promising matrix for aquafeed production [6,7]. Microalgae are renowned for their balanced biochemical composition, high nutritional value with an interesting content in proteins, long-chain polyunsaturated fatty acids, vitamins and minerals [8] that are suitable to support the nutritional needs of fish [6]. Moreover, the presence of bioactive molecules, such as carotenoids, showed nutraceutical and immunostimulant properties, highlighted in various studies in different fish species [9–11]. To date, high production cost and limited availability are the main limiting factors for the use of algal biomass in aquaculture.

In this context, a circular bioeconomy approach for the conversion of sustainable resources into bio-based products and bioenergy, including food and feed, using innovative and efficient industrial biotechnologies, is an opportunity to improve ethic and environmental sustainability and to optimize production processes and lower costs.

Recent research shows a good ability of different microalgal species to grow on complex waste stream of organic origin (digestate, wastewater, etc.) performing the remediation of nutrient pollutants (P, N, etc.), while producing biomass [12,13]. Therefore, the development of integrated biorefineries based on the use of microalgal biotechnology would allow the recovery of some agro-industrial by-products reducing disposal costs.

The microalgae biomass produced from biorefineries could be a valuable source of proteins, lipids and used as bioactive compounds for aquafeed but, at present, very few studies have been conducted on this topic. In particular, few studies conducted on Acipenserides showed that feed supplemented with *Arthrospira* sp. and vegetable oils has proved to be a valid alternative to fish oil and fish meal, both in white (*A. transmontanus*) and Siberian (*A. baerii*) sturgeon diet [14].

The research activities carried out in the current study aimed to assess the partial replacement of FM and FO in the aquafeed with a blend of two microalgae, *Scenedesmus* sp. *and Chroococcus* sp. deriving from an integrated biorefinery. The quality of the biomass and the experimental diet was assessed by a detailed biochemical and microbiological characterization and by a feeding trial on Siberian sturgeon fingerlings (*Acipenser baerii*).

## 2. Materials and Methods

### 2.1. Microalgae Biomass Production

The experimentation lasted 200 days and was hosted at a farm breeding approximately 20,000 pigs located in Northern Italy. At the farm, a mixture of wastes from farming and agro-practices were processed in an anaerobic digester to produce bioenergy via anaerobic digestion. The output of the biogas plant, namely the digestate, was sent to a centrifuge to produce a clarified liquid known as 'centrate'; the centrate showed a variable composition, but a generally high pollution load ($630 \pm 330$ mg $N\text{-}NH_4^+$ $L^{-1}$, $13 \pm 6$ mg $P\text{-}PO_4^{3-}$ $L^{-1}$, $1090 \pm 380$ mg COD $L^{-1}$), a turbidity of $190 \pm 170$ FAU and a conductivity of $9.2 \pm 3.9$ mS $cm^{-1}$. The centrate was used to feed microalgae/bacteria grown in a pilot-scale raceway pond (RW). The RW consisted of a raceway

with a surface of approximately 3.8 m$^2$ and volume of 0.50–0.88 m$^3$, depending on the water depth. The pond was mixed by a paddlewheel and featured a sump for pH-controlled $CO_2$ dosage; more details about the experimental set-up are available in Pizzera et al. [13]. Biomass harvesting was performed with a pilot-scale centrifuge (10,000 *g* from Filtermaxx, USA), and the liquid fraction was partially or totally recirculated to the RW. Therefore, the hydraulic retention time (HRT) and the biomass retention time (SRT) were uncoupled (on average HRT = 60 d, SRT = 35 d). The harvested biomass was air dried at 32 °C.

During the first 7 weeks of start-up, the RW was fed on diluted centrate to favor microalgae adaptation; afterwards, the plant was operated for 150 days by feeding undiluted centrate. The pilot was monitored in order to assess the bioremediation performances as for the organic matter and nutrient (N and P) content. Moreover, the concentration of dry matter (DM) and the cell counts in the algal suspension were assessed to monitor the density and composition of the microbial community. Details on analytical methods and data processing procedures are reported elsewhere [13]. The climate was typical of Northern Italy (average daily temperature from 3 to 30 °C, daily precipitation from 0 to 59 mm). High evaporation occurred (9 ± 7 L·d$^{-1}$) with respect to incoming flow (from 0 to 49 L·d$^{-1}$).

## 2.2. Feed Formulation and Production

Two experimental diets were formulated to be isoproteic (51.0% DM) and isolipidic (14.0% DM). The control diet (CT) mimics the ingredient composition of the commercial microalgae-free diets used for feeding sturgeon fingerlings. The diet SC 10 was formulated including 10% (w/w) of a blend *Scenedesmus-Chroococcus*, partially replacing FM. The composition of ingredients of the diets is shown in Table 1. Formula was also established on the base of the nutritional composition the dried algal biomass (mainly consisting of a blend of *Scenedesmus-Chroococcus*) (Table 2).

**Table 1.** Ingredient composition of experimental diets used in the feeding trial.

| *Ingredients* (g kg$^{-1}$ Dry Matter, DM) | CT | SC 10 |
|---|---|---|
| Fish meal [1] | 274.0 | 191.0 |
| Soybean protein concentrate [2] | 250.0 | 250.0 |
| Gluten meal [3] | 150.0 | 150.0 |
| Dried *Scenedesmus-Chroococcus* | 0 | **100.0** |
| Attractant premix [4] | 100.0 | 100.0 |
| Fish solubles CPSP 90 [5] | 50.0 | 50.0 |
| Fish oil | 50.0 | 53.4 |
| Wheat meal [6] | 40.5 | 19.5 |
| Soybean oil | 27.0 | 27.6 |
| Soybean lecithin [7] | 10.0 | 10.0 |
| Vitamin and mineral premix [8] | 10.0 | 10.0 |
| Guar gum | 10.0 | 10.0 |
| Alginate | 10.0 | 10.0 |
| Choline chloride [7] | 5.0 | 5.0 |
| Betaine | 5.0 | 5.0 |
| Lysine | 5.0 | 5.0 |
| Methionine | 2.5 | 2.5 |
| Stay C Roche 0.2% | 1.0 | 1.0 |

[1] (protein: 69.4%; lipids: 12.3%), Norsildemel (Bergen, Noruega); [2] (protein: 51.5%; lipids: 8.0%); [3] (protein: 76.0%; lipids: 1.9%); [4] (50% squid meal, 50% krill meal); [5] (protein: 84.1%; lipids: 8.8%), Sopropeche (France); [6] (protein: 12.0%; lipids: 2.0%); [7] SigmaAldrich (Madrid, Spain); [8] Vitamin and mineral premix.

For the feed production, the dry microalgal biomass was provided to the Ceimar-University of Almería (Service of Experimental Diets), where all the ingredients were ground and mixed in a vertical spiral-shaped mixer (Sammic BM-10, capacity 10-L, Sammic, Azpeitia, Spain) and subsequently supplemented with fish oil and diluted choline. The ingredients were mixed for 15 min, then integrated with water to obtain a homogeneous dough that was subjected to a cold extrusion process (Miltenz 51SP,

JSConwell Ltd. New Zealand) for making pellets with 1 mm diameter and 1.5 mm length. A temperature of about 60 °C was applied for the extrusion process for preserving the potential functional properties of the microalgae. The pellets were dried in a chamber at 25 °C (Air-Frio, Almería, Spain). After 24 h, the feed was stored, in plastic bags under vacuum packaging conditions, at −20 °C until use.

### 2.3. Fish, Feeding Trial and Sampling

The feeding trial was carried out at the experimental facilities of the Italian Experimental Institute Lazzaro Spallanzani (Rivolta d'Adda, Italy). Six groups of Siberian sturgeon (*A. baerii*) fingerlings (mean body weight 12.0 ± 0.1 g), each consisting of 16 individuals, were randomly sampled from a single group of 96 fish and kept in six 120 L fiberglass tanks in RAS system (Recirculating Aquaculture System, daily water exchange, 2%, with mechanically filtered and UV-treated water).

Water parameters were monitored daily and kept constant and optimal for this species (temperature 18.9 ± 0.6 °C, dissolved oxygen 9.6 ± 1.2 mg L$^{-1}$, pH 8.1 ± 0.1, NH$_4$-N < 0.06 mg L$^{-1}$, NO$_2$-N < 0.2 mg L$^{-1}$). The photoperiod used was 12 h of artificial light and 12 h of darkness. After 15 days of acclimatization, dietary treatments were randomly assigned in triplicate to the 6 groups. Fish were fed for 40 days with experimental diets by hand, six days per week in two daily meals (9:00 a.m. and 5:00 p.m.) with a feed ratio equal to 3% of body weight. At days 15, 30 and at the end of the growth trial (40 days), fish were group-weighed after a 24 h fasting period, under moderate anesthesia (MS222, 50 mg L$^{-1}$) to assess zootechnical parameters.

Growth performance and nutrient utilization were estimated using the following parameters: survival rate (SR, %), initial body weight (IBW, g), final body weight (FBW, g), total feed intake (FI), specific growth rate (SGR) and feed conversion ratio (FCR) were computed as shown below:

SGR (%): 100 × [(ln final body weight − ln initial body weight)/days]

FCR: Total feed intake (g)/weight gain (g)

Six fish per tank (18 fish per dietary treatment) were randomly selected, euthanized with a bath of tricaine (Pharmaq) at a lethal concentration, then dried on absorbent paper and subjected to individual biometric measurements (total length, body weight). The condition index was then calculated following Fulton's K-index:

K = (W/L$^3$) × 100, where W is the weight and L is the total length of the fish.

A pool of the respective skinned fillets from 3 fish per tank (9 fish per dietary treatment) was frozen and stored at −20 °C for proximate and fatty acid analysis. A pool of livers from the same fish was frozen and stored at −20 °C until superoxide dismutase (SOD) and catalase (CAT) activity were measured. In addition, 2 fish per tank (6 fish per dietary treatment) were sacrificed and dissected for intestinal tract sampling and subsequent enzymatic, TEM and SEM microscopy analysis.

In this study, EU directive 2010/63/EU and the relative guidelines on the protection of animals used for scientific purposes were followed for fish manipulation.

### 2.4. Chemical Analysis

The dried microalgae biomass, experimental diets and fillet muscle tissue were analyzed for dry matter, crude protein, total lipids and ash according to AOAC methods [15], after grinding and homogenization.

The Folch method [16] with a chloroform–methanol mixture (2:1, *v/v*), was used for the extraction and determination of total lipids. The preparation of fatty acid methyl esters (FAME) was performed according to Christie [17].

Amino acids analysis was performed using 100 mg of sample, which was hydrolyzed in 10 mL of 6 N HCl under vacuum at 110 °C for 24 h. After hydrolyzation, the sample was filtered (using filter with pores of 0.45 μm), washed with 3 × 1 mL of 0.1 N HCl, evaporated to dryness under nitrogen at 40 °C, and the dry residue was redissolved in 2 mL of distilled water. The Graser et al. [18] method was used for amino acid determination. The HPLC consisted of a Perkin Elmer Series 200 pump, equipped with a Perkin Elmer Series 200 Autosampler and a Perkin Elmer Altus A-10 FL Detector.

The gradient elution was used for the separation. Mobile phase A consisted of a methanol/acetonitrile solution in a 12:1 (*v/v*) ratio, mobile phase B was 23 mM sodium acetate pH 5.95 [19]. After applying a linear gradient for 75 min at a flow rate of 1 mL min$^{-1}$ from 0% to 53.5% B, an equilibration step was performed with 100% A for 20 min [20].

### 2.5. Microbiological Analysis

A microbiological characterization of the dried algal biomass and the formulated experimental diets were carried out before their use in the feeding trial. Each sample (10 g) was homogenized in 90 mL of sterile 0.1% peptone water solution (PWS, Oxoid™ Peptone Bacteriological, Thermo Fisher Diagnostics, Rodano, Italy) using the homogenizer Stomacher 400 circulator (VWR International, Milan, Italy). The homogenized samples were ten-fold diluted in PWS and plated by inclusion in agar media for specific bacterial counts, as follows: (i) total mesophilic aerobic bacteria on Plate Count Agar (Oxoid) for 72 h at 30 °C [21]; (ii) Enterobacteriaceae on Violet Red Bile Glucose agar (Oxoid) for 24 h at 37 °C [22]; *Escherichia coli* on ChromID Coli agar (Biomerieux, Bagno a Ripoli, Italy) for 24 h at 44 °C [23]; Enterococci on Kanamycin Esculine Azide agar (Merck, Darmstadt, Germany) for 24–48 h at 42 °C. Sulfite-reducing Clostridia spores were enumerated after heat treatment of each homogenized sample to 80 °C for 10 min. Sample dilutions were inoculated in TSC (Tryptose Sulfite Cycloserine agar, Merck) and incubated for 24–48 h at 37 °C under anaerobic conditions [24]. The presence of *Clostridium perfringens* spores was confirmed by testing typical colonies on TSC agar for acid phosphatase activity according to ISO standard method [25]. To detect acid phosphatase, a commercial kit was used according to manufacturer instructions (Acid phosphatase reagent TN1519, Sifin Diagnostics, Berlin, Germany). The presence/absence of *Salmonella* spp. in 25 g of each sample was determined according to ISO standard 6579-1 [26]. The microbiological determinations were carried out in duplicate. The bacterial count data were log-transformed, and results were expressed as means (log CFU g$^{-1}$) ± standard deviations (SD).

### 2.6. SOD, CAT Analysis

Before performing enzymatic activity assays, an aliquot of sturgeon livers from the pool (about 50–100 mg from each fish) was rinsed with a phosphate-buffered saline (PBS) solution, pH 7.4, to remove any red blood cells and clots. The tissue was then homogenized on ice in 10 mL g$^{-1}$ tissue of cold buffer (50 mM potassium phosphate, pH 7.0, containing 1 mM EDTA for CAT assay and 20 mM HEPES, pH 7.2, 1 mM EGTA, 210 mM mannitol and 70 mM sucrose for SOD assay). Tissues homogenized with respective buffers were centrifuged at 10,000× *g* for 15 min, at 4 °C and the supernatant was stored at −80 °C until the assays were performed. Catalase Assay Kit (Item No. 707002) and Superoxide Dismutase Assay Kit (Item No. 706002) were purchased from Cayman Chemical (Ann Arbor, Michigan). Absorbance was read using an Infinite®F500 (Tecan Trading AG, Switzerland) spectrometer. Each sample was tested in triplicate.

The CAT measurement was based on the reaction of the enzyme CAT with methanol in the presence of an optimal concentration of $H_2O_2$. The formaldehyde produced by the oxidation step was measured colorimetrically using Purpald (4-Amino-3-hydrazino-5-mercapto-1,2,4-triazole, 4-Amino-5-hydrazino-1,2,4-triazole-3-thiol) as chromogen. The absorbance was read at 540 nm. CAT activity was expressed as the amount of enzyme that caused the formation of 1.0 nmol of formaldehyde (oxidation product) per min at 25 °C (nmol min$^{-1}$ mL$^{-1}$). Quantification was performed by a calibration curve, obtained by plotting the absorbance of a known standard as a function of the final formaldehyde concentration.

For SOD determination, a tetrazolium salt was used as radical detector of superoxide radicals generated by xanthine oxidase and hypoxanthine. Absorbance of samples having undergone oxidation was read at 450 nm. Total SOD activity (cytosolic and mitochondrial) was expressed as unit of enzyme that exhibits 50% dismutation of the superoxide radical (U mL$^{-1}$). Quantification was performed

plotting the SOD linearized standard rate, obtained dividing the absorbance of a known standard for the absorbance of samples, as a function of final SOD activity.

### 2.7. Analysis of Digestive Enzyme Activities

For the enzymatic activity determination, proximal and distal intestines from animals of each group were separately homogenized in ice-cold distilled water (1:2 *w/v*). Supernatants were obtained following centrifugation at 13,000× *g* during 12 min at 4 °C, and then stored at −20 °C until analysis. The activity of total alkaline proteases was determined at 280 nm according to Alarcón et al. [27], using buffered 5 g L$^{-1}$ casein (50 mM Tris HCl pH 9.0). The amount of enzyme releasing 1 µg tyrosine min$^{-1}$ was defined as one unit of activity (extinction coefficient of tyrosine 0.008 µg$^{-1}$ mL$^{-1}$ cm$^{-1}$). The activities of trypsin and chymotrypsin were measured with 0.5 mM BAPNA (N-a-benzoyl-DL-arginine-4-nitroanilide) following the methodology of Erlanger et al. [28] and 0.2 mM SAPNA (N-succinyl-(Ala)$_2$-Pro-Phe-P-nitroanilide) using the procedure described by DelMar et al. [29], respectively. The activity of leucine aminopeptidase was assayed using buffered 2 mM L-Leucine-p-nitroanilide (pH 8.8) [30]. The alkaline phosphatase was quantified using buffered p-nitrophenyl phosphate (pH 9.5) [31]. One unit of trypsin, chymotrypsin and leucine aminopeptidase activity was established as the amount of enzyme that released 1 µmol p-nitroanilide min$^{-1}$ (extinction coefficient 8800 M cm$^{-1}$ at 405 nm). In the case of alkaline phosphatase, a unit of activity was defined as the enzyme amount that releases 1 µg nitrophenyl min$^{-1}$ (extinction coefficient 17,800 M cm$^{-1}$ at 405 nm). The specific enzymatic activity was expressed as U g tissue$^{-1}$. All the assays were performed in triplicate. In the case of total alkaline proteases, substrate-SDS-PAGE electrophoresis gels were performed for visualizing those enzymes.

### 2.8. Electron Microscopy Study of the Fish Intestinal Mucosa

Proximal and distal intestine samples for transmission electron microscopy (TEM) were treated for 4 h at 4 °C with buffered 2 g L$^{-1}$ glutaraldehyde and 40 g L$^{-1}$ formaldehyde (pH 7.5). Then, tissue sections were washed with PBS (three times for 20 min). After that, samples were treated with 20 g L$^{-1}$ osmium tetroxide and were afterwards dehydrated in gradient ethanol solutions ranging from 50% to 100% (*v/v*). Samples were embedded in ethanol 100% (*v/v*) and Epon resin (1:1) for 2 h under continuous shaking. Next, samples were included in Epon resin during 24 h at 60 °C. Ultra fine cuts were obtained and then placed on a 700 Å cooper mesh and stained with lead citrate and uranyl acetate. The samples were visualized using a transmission electron microscope (Zeiss 10 C at 100 Kv, Carl Zeiss, Barcelona, Spain). TEM micrographs were analyzed using UTHSCA ImageTool software for measuring microvilli length and microvilli diameter, and the number of microvilli over 1 µm$^2$ surface [32]. One hundred measurements of the proximal and distal intestine per dietary group were carried out. Total absorption surface per microvilli was determined according to Vizcaíno et al. [9]. Additional samples were obtained and processed for observation in a scanning electron microscopy (SEM) (HITACHI model S-3500, Hitachi High-Technologies Corporation, Japan) according to Vizcaíno et al. [9].

### 2.9. Statistical Analysis

Data of growth performance, biometric indices, proximate composition, fatty acids, antioxidant enzymes are expressed as mean ± standard deviation. Prior to statistical analysis, all the data were evaluated for normality distribution. Means were compared using the Student's *t*-test, set for *p* < 0.05. All analyses were carried out using the SPSS-PC release 17.0 (SPSS Inc., Chicago, IL, USA).

## 3. Results and Discussion

### 3.1. Microalgae Biomass Production

Despite some suboptimal characteristics of the undiluted centrate (high ammonium concentration, high N/P ratio, high turbidity), the microalgae consortium grew stably and the algal biomass concentration was very high for this kind of reactor, being on average 1.2 ± 0.6 g DM L$^{-1}$. This high

concentration was also achieved thanks to the high evaporation, which led to a concentration effect. Average algal productivity was $6.2 \pm 1.0$ g DM·m$^{-2}$·d$^{-1}$ during the operation with undiluted centrate, which is coherent with results reported in published experimentations on similar systems [13,33,34]. As for the bioremediation performances, removal efficiencies were computed by comparing the influent and effluent loads for relevant pollutants (details on data processing can be found in Pizzera et al. [13]) and were on average $99.7 \pm 0.7\%$ for ammonium, $86 \pm 15\%$ for phosphorus, and $57 \pm 15\%$ for organic matter (assessed in terms of chemical oxygen demand, COD). These data demonstrate the algae/bacteria consortia were effective in reducing the pollution load in the centrate and could be applied to reduce the environmental impact of intensive farming, especially in those areas that are classified as sensitive to nitrate according to the European Nitrite Directive (91/676/CEE). The microalgal community was mixed and mainly made up of green microalgae belonging to the Chlorellaceae and Scenedesmaceae families, as well as of cyanobacteria of the Chroococcaceae family with total algal count varying within 0.5–9.0 10$^6$ cell mL$^{-1}$. From July onward, branched green algae belonging to the Chaetophoraceae family became dominant (*Stigeoclonium*). The biomass used for this study was collected from the pilot plant with four samples from 22/06 to 02/07/2018, therefore it was mainly composed of *Scenedesmus* sp. and *Chroococcus* sp. (Figure 1).

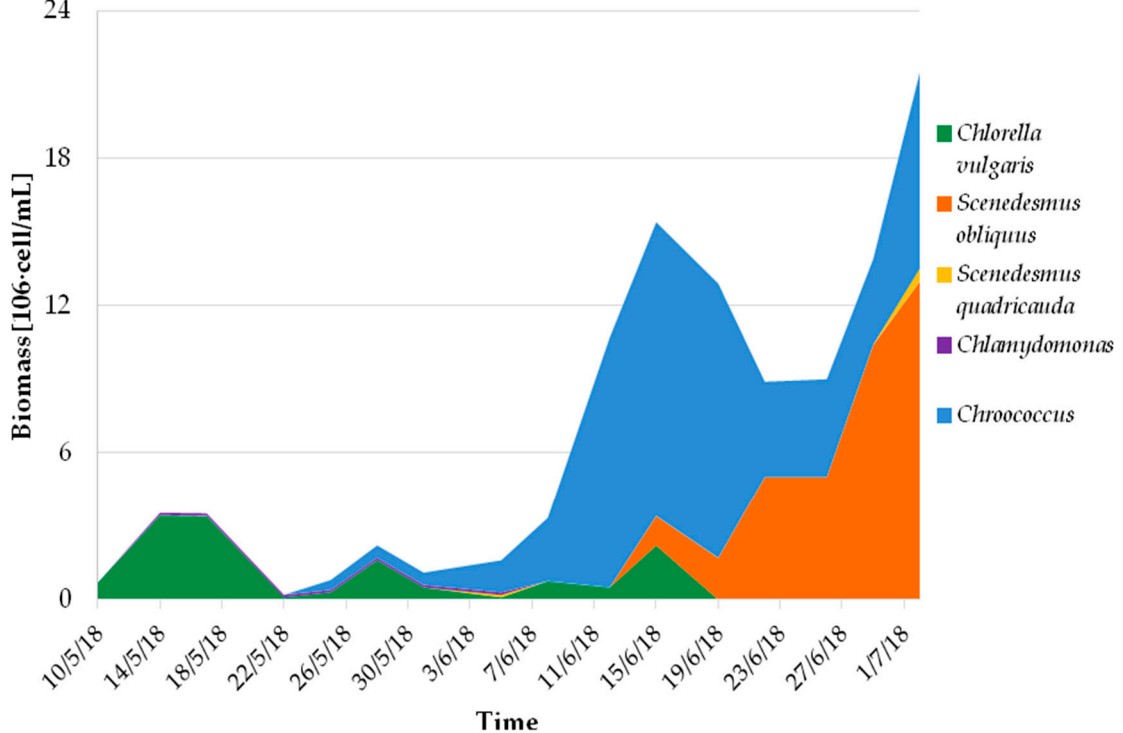

**Figure 1.** Growth and evolution of the community composition of microalgae culture used for the elaboration of experimental diet.

### 3.2. Microalgae Biomass Chemical and Microbiological Characterization

The chemical composition and microbiological traits of the dried biomass of *Scenedesmus* and *Chroococcus* grown on digestate are shown in Tables 2 and 3, respectively. Microbial contamination is an important issue when microalgae are produced in open-air systems [35], especially when a non-sterile substrate is used. The freeze-dried biomass of microalgal community cultivated on digestate showed a total bacterial content of 5.98 log CFU g$^{-1}$, with Enterobacteriaceae and *E. coli* at 2.73 and 2.49 log CFU g$^{-1}$, respectively. Sulfite-reducing Clostridia spores were found at level of about 4.03 log CFU g$^{-1}$ and characteristic colonies were confirmed as *C. perfringens* by acid phosphatase test [36]; *Salmonella* spp. was not found in 25 g of dried biomass.

**Table 2.** Proximate composition, amino acid (AA) and fatty acid profile of *Scenedesmus–Chroococcus* air dried biomass (data expressed on dry matter basis).

| Proximate Composition (g/100 g Microalgae) | *Scenedesmus–Chroococcus* Blend |
|---|---|
| Water | 10.1 |
| Crude protein | 47.4 |
| Total lipids | 9.6 |
| Carbohydrates | 9.7 |
| Ash | 6.8 |
| **Essential AA (g/100 g Microalgae)** | ***Scenedesmus–Chroococcus* Blend** |
| Arginine | 6.00 |
| Histidine | 2.18 |
| Isoleucine | 2.27 |
| Leucine | 1.55 |
| Lysine | 3.42 |
| Methionine | 1.46 |
| Phenylalanine | 1.28 |
| Tyrosine | 1.27 |
| Threonine | 2.23 |
| Valine | 1.17 |
| **Non Essential AA (g/100 g Microalgae)** | ***Scenedesmus–Chroococcus* Blend** |
| Alanine | 0.76 |
| Aspartic acid | 8.34 |
| Glutamic acid | 5.60 |
| Glycine | 0.88 |
| Serine | 2.66 |
| **Fatty Acid Composition (mg/100 g Fatty Acid)** | ***Scenedesmus–Chroococcus* Blend** |
| 14:0 | 0.69 |
| 16:0 | 21.59 |
| 16:1n−7 | 1.62 |
| 17:0 | 3.96 |
| 16:2n-4 | 0.48 |
| 18:0 | 13.15 |
| 18:1n-9 | 10.29 |
| 18:1n-7 | 1.20 |
| 18:2n-6 | 16.61 |
| 18:3n-3 | 25.82 |
| 18:4n-3 | 4.59 |
| **SFA** | 39.38 |
| **MUFA** | 13.11 |
| **PUFA** | 47.50 |
| **n-3** | 30.41 |
| **n-6** | 16.61 |
| **n-3/n-6** | 1.83 |

Abbreviations: SFA, saturated fatty acids; MUFA, monounsaturated fatty acids; PUFA, polyunsaturated fatty acids.

**Table 3.** Microbial content of *Scenedesmus–Chroococcus* air dried biomass.

| Indicator Parameters | *Scenedesmus-Chroococcus* Blend |
|---|---|
| Total viable aerobic count (log CFU $g^{-1}$) | 5.98 ± 0.03 |
| Enterobacteriaceae (log CFU $g^{-1}$) | 2.73 ± 0.07 |
| *E. coli* (log CFU $g^{-1}$) | 2.49 ± 0.20 |
| *Salmonella* spp. (in 25 g) | absent |
| Sulfite-reducing Clostridia spores (log CFU $g^{-1}$) | 4.03 ± 0.02 * |

Mean values ± standard deviations. * presence of *C. perfringens* colonies confirmed by positive reaction to acid phosphatase assay.

### 3.3. Chemical Composition and Microbiological Characterization of Experimental Diets

The chemical composition of the diets is shown in Table 4. The dry matter, crude protein, total lipids, ash contents were similar between the diets, and adequate to fulfill the nutritional and energy requirements of Siberian sturgeon fingerlings.

**Table 4.** Proximate composition and fatty acid profile of the experimental diets.

| Proximate Composition (g/100 g Feed on Dry Basis) | CT | SC 10 |
|---|---|---|
| Moisture | 11.1 | 10.9 |
| Crude protein | 51.5 | 51.6 |
| Crude lipid | 14.0 | 14.3 |
| Ash | 7.6 | 7.6 |
| **Fatty Acid Composition (g/100 Fatty Acid)** | **CT** | **SC 10** |
| 14:0 | 4.42 | 4.46 |
| 15:0 | 0.38 | 0.40 |
| 16:0 | 17.08 | 17.92 |
| 16:1n−7 | 3.83 | 3.88 |
| 17:0 | 0.28 | 0.40 |
| 16:2n-4 | 0.40 | 0.38 |
| 16:3n-4 | 0.39 | 0.38 |
| 18:0 | 3.19 | 3.22 |
| 18:1n-9 | 15.97 | 15.99 |
| 18:1n-7 | 2.41 | 2.39 |
| 18:2n-6, LOA | 20.42 | 20.92 |
| 18:3n-3 | 2.39 | 2.90 |
| 20:0 | 0.24 | 0.26 |
| 18:4n-3 | 1.73 | 1.82 |
| 20:1n-11 | 0.33 | 0.30 |
| 20:1n-9 | 3.52 | 3.09 |
| 20:4n-6, ARA | 0.54 | 0.52 |
| 22:1n-11 | 4.71 | 4.25 |
| 22:1n-9 | 0.40 | 0.35 |
| 20:5n-3, EPA | 7.42 | 6.97 |
| 24:1 | 0.48 | 0.45 |
| 22:5n-3 | 0.75 | 0.75 |
| 22:6n-3, DHA | 8.74 | 7.97 |
| **SFA** | 25.59 | 26.66 |
| **MUFA** | 31.63 | 30.71 |
| **PUFA** | 42.78 | 42.63 |
| **n-3** | 21.04 | 20.42 |
| **n-6** | 20.96 | 21.45 |
| **n-3/n-6** | 1.00 | 0.95 |

Abbreviations: LOA, linoleic acid; EPA, eicosapentaenoic acid; ARA, arachidonic acid; DHA, docosahexaenoic acid; SFA, saturated fatty acids; MUFA, monounsaturated fatty acids; PUFA, polyunsaturated fatty acids.

Regarding the fatty acid profile, overall, no differences were observed between experimental diets. The level of saturated fatty acids (SFA) ranged between 25.6% and 26.7%, respectively, in the control and microalgae-supplemented diets. Monounsaturated fatty acids (MUFA) were mainly represented by the oleic acid C18:1n-9, ranging between 30.7% (SC 10) and 31.6% (CT). Polyunsaturated fatty acids (PUFA), with values of 42.8% (CT) and 42.6% (SC 10), was the most abundant category of fatty acids, and linoleic acid (18:2n-6) with values of 20.4% (CT) and 20.9% (SC 10) was mainly represented.

Eicosapentaenoic acid (EPA) was present, ranging from 6.97 to 7.42% in both diets, while docosahexaenoic acid (DHA) ranged from 7.97 to 8.7% in the experimental and control diets, respectively.

Microbiological traits of the test diets are shown in Table 5. Both the control and experimental diet showed a lower microbial load, probably due to the dilution of the experimental ingredient and to the

heat treatment applied during the extrusion process. The total bacterial content was slightly higher in the experimental diet (4.36 vs. 3.69 log CFU g$^{-1}$), while Enterobacteriaceae, *E. coli* and *Salmonella* spp. were not detected.

**Table 5.** Microbial content values in the experimental diets.

| Indicator Parameters | CT | SC 10 |
|---|---|---|
| Total viable aerobic count (log CFU g$^{-1}$) | 3.69 ± 0.21 | 4.36 ± 0.17 |
| Enterobacteriaceae (log CFU g$^{-1}$) | <2.00 | <2.00 |
| *E. coli* (log CFU g$^{-1}$) | <2.00 | <2.00 |
| *Salmonella* spp. (in 25 g) | absent | absent |
| Sulfite-reducing Clostridia spores (log CFU g$^{-1}$) | <2.00 | 2.83 ± 0.03 * |

Mean values ± standard deviations. * presence of *C. perfringens* colonies confirmed by positive reaction to acid phosphatase assay.

A low number of sulphite-reducing Clostridia spores, including *C. perfringens*, were present only in the experimental diet, probably deriving from digestate. These microorganisms are obligate anaerobic bacteria, so they cannot survive in a microalgal culture, since they are rich in oxygen due to intense photosynthesis. The absence of *Salmonella*, considered the major hygienic risk for animal feed, ensures its safety [37]. The new fertilizing products regulation (UE n. 2019/1009) states that digestate, even coming from organic waste (except for sludge from civil wastewater), can be used for the production of fertilizers, if no *Salmonella* spp. is present in 25 g sample and *E. coli* is less than 1000 CFU/g of fresh mass.

Concerning microbiological feed quality, everything is demanded to voluntary quality standard, HACCP evaluation and the principles contained in the Codex Alimentarius (Regulation (EC) No 183/2005 laying down requirements for feed hygiene). In general, the use of centrate as by-product stream for microalgal cultivation, does not seem to particularly affect the microbiological quality of experimental diet.

*3.4. Growth Performance, Nutrient Utilization and Fillet Chemical Composition*

Both CT and SC 10 diets were palatable and accepted by fish. Mortality was 2% on average, and the dietary treatments did not affect any parameters analyzed. The growth of *A. baerii* fed on the experimental diet throughout the 40-day trial is shown in Figure 2.

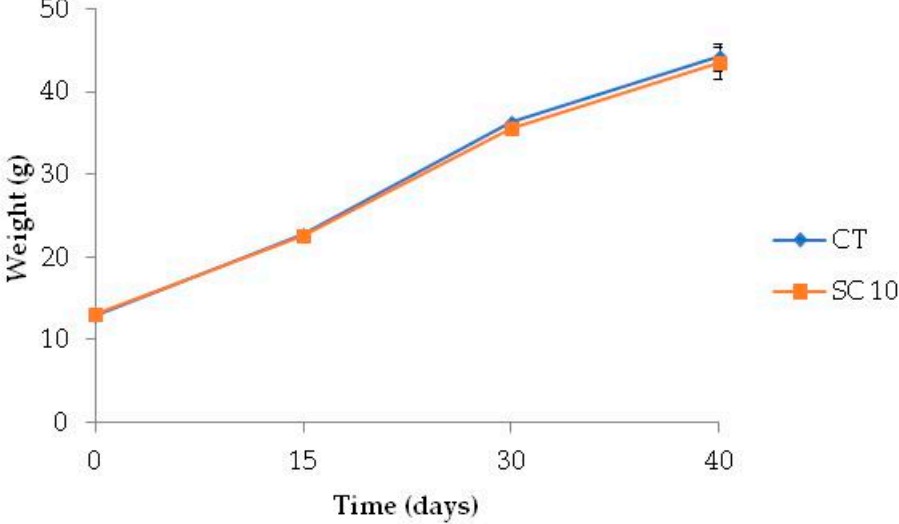

**Figure 2.** Time course variation of body weight of fish fed the different diets for 40 days. (*n* = 3, 16 fish per tank).

At the end of feeding period, the Final Body Weight (FBW), as well as the other growth and nutrient utilization parameters (FI, SGR, FCR) in fish fed microalgae-supplemented feed, did not differ from the group fed with CT diet, and K-factor and somatic indices were also similar between dietary treatments (Table 6).

**Table 6.** Growth performance, nutrient utilization, and somatic indices of *A. baerii* fingerlings fed the test diets over 40 days.

| Growth Parameters | CT | SC 10 |
|---|---|---|
| IBW (g) | 12.8 ± 0.2 | 13.0 ± 0.2 |
| FBW (g) | 44.2 ± 1.7 | 43.5 ± 2 |
| FI [1] (g) | 346.2 ± 3.7 | 341.5 ± 1.5 |
| SGR [2] | 3.1 ± 0.05 | 3.0 ± 0.08 |
| FCR [3] | 0.69 ± 0.02 | 0.70 ± 0.05 |
| Survival (%) | 100 ± 0.0 | 97.9 ± 4.4 |
| **Somatic Indices** | **CT** | **SC 10** |
| Total length (cm) | 25.2 ± 1.8 | 23.9 ± 1.9 |
| Whole body weight (g) | 38.6 ± 9.35 | 39.8 ± 8.6 |
| K-factor | 0.30 ± 0.02 | 0.29 ± 0.03 |

[1] FI: Feed intake (g feed ingested). [2] SGR:$100 \times [(\ln \text{final body weight} - \ln \text{initial body weight})/\text{days}]$. [3] FCR: feed intake/weight gain.

The growth performance of Siberian sturgeon used in this trial showed similar results to other feeding trial with the same species used during the same growing phase.

Yun et al. [38], in a trial with the total replacement of fish meal with plant blend meal, observed the growth of sturgeon from 39 g to 135–144 g in an eight-week trial, with a SGR ranging from 2.21 to 2.33 according to the different diet tested, while Caimi et al. [39] used juvenile Siberian sturgeon weighting 24 g in a trial where they tested *Hermetia illucens* meal at two level of inclusion and a vegetable-based diet. After 118 days of trial, fish showed a final weight ranging from 141 to 159 g with a SGR of 1.58–1.59 according to the diet that they had received.

The inclusion of microalgae did not affect the fillet composition in terms of total protein, lipid and ash content in *A. baerii* fingerlings (Table 7).

Concerning the fatty acid profile, the fillet composition mirrored the composition of the diets, with a predominance of PUFA followed by MUFA and SFA. Considering single fatty acids, the fatty acids present in higher amount were oleic acid, palmitic acid and linoleic acid, which were the same prevalent fatty acids found in the two diets tested. Sturgeon fed with SC 10 diet showed a lower SFA content ($p < 0.05$), partially due to a minor content of palmitic acid (16:0), even if the SC 10 diet was not the poorest regarding this fatty acid in comparison with the control diet. Other differences highlighted in Table 7 reflect the minimal differences in the fatty acid composition of the two diets.

**Table 7.** Proximate composition and fatty acid profile of fillets of *A. baerii* fingerlings fed the test diets over 40 days.

| Proximate Composition (g/100 g Muscle on Wet Basis) | CT | SC 10 | |
|---|---|---|---|
| Moisture | 79.05 ± 1.73 | 79.34 ± 0.26 | |
| Total protein | 16.55 ± 1.10 | 16.63± 0.62 | |
| Total lipids | 3.48 ± 1.06 | 3.13 ± 0.45 | |
| Ash | 0.92 ± 0.11 | 0.90 ± 0.09 | |
| **Fatty Acid Composition (g/100 g Fatty Acids)** | | | *p* |
| 14:0 | 3.37 ± 0.19 | 3.47 ± 0.11 | |
| 16:0 | 18.96 ± 0.28 | 18.08 ± 0.07 | * |
| 16:1n−7 | 3.67 ± 0.17 | 4.00 ± 0.19 | |
| 18:0 | 3.35 ± 0.24 | 3.09 ± 0.29 | |

**Table 7.** *Cont.*

| Proximate Composition (g/100 g Muscle on Wet Basis) | CT | SC 10 | |
|---|---|---|---|
| 18:1n-9 cis | 19.73 ± 0.37 | 19.99 ± 0.57 | |
| 18:1n−7 | 3.24 ± 0.10 | 3.23 ± 0.10 | |
| 18:2n-6 cis 9.12 | 17.04 ± 0.27 | 17.11 ± 0.37 | |
| 18:3n-6 | 0.39 ± 0.10 | 0.66 ± 0.10 | * |
| 18:3n-3 | 1.64 ± 0.09 | 1.96 ± 0.09 | * |
| 18:4n-3 | 0.90 ± 0.09 | 1.02 ± 0.09 | |
| 20:1n-11 | 1.18 ± 0.02 | 0.96 ± 0.02 | |
| 20:1n-9 | 3.53 ± 0.02 | 3.21 ± 0.08 | * |
| 20:3n-6 | 0.33 ± 0.02 | 0.38 ± 0.03 | |
| 20:4n-6 | 0.92 ± 0.11 | 0.95 ± 0.08 | |
| 22:1n-11 | 2.72 ± 0.14 | 2.39 ± 0.16 | ** |
| 20:5n-3 | 5.40 ± 0.07 | 5.57 ± 0.07 | * |
| 22:5n-3 | 1.66 ± 0.14 | 1.84 ± 0.07 | |
| 22:6n-3 | 11.99 ± 0.92 | 12.10 ± 0.53 | |
| **SFA** | 25.68 ± 0.22 | 24.63 ± 0.17 | ** |
| **MUFA** | 34.06 ± 0.54 | 33.78 ± 0.49 | |
| **PUFA** | 40.26 ± 0.74 | 41.58 ± 0.36 | |
| **n-3** | 21.59 ± 0.95 | 22.49 ± 0.48 | |
| **n-6** | 18.67 ± 0.24 | 19.10 ± 0.34 | |
| **n-3/n-6** | 1.16 ± 0.06 | 1.18 ± 0.04 | |

Abbreviations: SFA, saturated fatty acids; MUFA, monounsaturated fatty acids; PUFA, polyunsaturated fatty acids.
* $p < 0.05$; ** $p < 0.001$.

### 3.5. Liver Antioxidant SOD, CAT Activity

No significant differences were detected in the enzymatic antioxidant activity in the liver of fish analyzed in this study (Table 8). Antioxidant enzymes are considered a primary defense line against the formation of toxic oxygen reactive species (ROS), using reactive species as substrates, thus performing direct detoxification [40]. Particularly, catalase (CAT, EC 1.11.1.6) is a ubiquitous antioxidant enzyme, present in most aerobic cells that destroys hydrogen peroxide to molecular oxygen and water, whether superoxide anion radical are converted to hydrogen peroxide and molecular oxygen by superoxide dismutase (SOD, EC 1.15.1.1) [41]. These enzymes were previously detected in several fish tissues, such as liver, kidney, gills, muscle, blood [42,43]. Several authors found that the activity of these enzymes varies in function of fish age [44], maturation stage [45], diet type [46], and some environmental factors, such as metabolic intensity and water temperature [47]. Rueda-Jasso et al. [46], demonstrated that high lipid dietary content induced an increase in the activity of CAT and SOD in liver. The same authors showed that the type of dietary starch also influences the level of the two enzymes since it affects the susceptibility of fish to oxidation. In the present study, no difference was observed between CAT and SOD activity in liver of *A. baerii* fingerlings fed with the control (CT) or the microalgae-supplemented (SC 10) diets. It was expected that fish fed with different diets managed the oxidative stress in different ways. This fact was previously confirmed when comparing high-lipid with low-lipid diets [46]. However, in this study, no difference in antioxidant activity in liver of fish from the two groups was found since the two feed formulations were isoproteic and isolipidic. Moreover, it is well known that another factor contributing to the oxidative status in fish is the dietary content of polyunsaturated fatty acids (PUFA) which are susceptible to peroxidation in the diets and when them are ingested and metabolized by fish [48]. In this study CT and the SC 10 diets did not show any significant difference in their fatty acid profile (Table 4) and, specifically, in their content in PUFA on the total of fatty acids was similar (42.78 and 42.63% in CT and SC 10, respectively). These results support the hypothesis that the two diets were comparable even when considering the antioxidant response induced in fingerlings of *A. baerii*. The microalgae biomass included in the experimental diet did not produce remarkable changes in the overall fatty acid composition of the diet and did not decrease the antioxidant activity against the toxic ROS in fish.

**Table 8.** Catalase (CAT) and superoxide dismutase (SOD) activity in liver of *A. baerii* fingerlings fed the test diets over 40 days.

| Diet | CAT Activity (nmol min$^{-1}$ 100 mg$^{-1}$) * | SOD Activity (U 100 mg$^{-1}$) ** |
|------|------|------|
| CT | 3.63 ± 0.32 | 0.188 ± 0.18 |
| SC 10 | 4.10 ± 0.58 | 0.154 ± 0.040 |

* = nmol of formaldehyde formed by CAT per minute in 100 mg of liver; ** = 1U is defined as the amount of SOD needed to exhibit 50% of dismutation of the superoxide radical in 100 mg of liver.

### 3.6. Digestive Enzyme Activity

Enzyme activities in proximal and distal intestine extracts of Siberian sturgeon specimens fed the experimental diets are shown in Table 9. The inclusion of microalgae significantly increased total proteolytic activity in proximal and distal intestine when compared to fish fed on a CT diet. An increment in digestive proteases could result in a better bioavailability of amino acids due to the hydrolysis of dietary protein by these enzymes. Trypsin activity level was similar between both groups, while chymotrypsin was significantly lower in the distal segment of fish fed SC 10 diet. Alkaline phosphatase and leucine aminopeptidase are brush border enzymes that play a relevant role in the final stages of digestive processes of dietary protein, allowing the absorption or transport of amino acids by enterocytes [49]. Regarding leucine aminopeptidase activity, fish fed on SC 10 showed significantly lower activity values, which might reflect a reduced capacity for delivering amino acids from the dietary protein. However, that eventual delay in the delivery rate of free amino acids from dietary protein as a consequence of reduced aminopeptidase activities seems to be compensated by efficient uptake and transport processes through the intestinal mucosa [9,49]. In this regard, alkaline phosphatase activity measured in both groups was similar and reflected the existence of efficient absorptive processes.

**Table 9.** Digestive enzyme activity (U g tissue$^{-1}$) in proximal (PI) and distal (DI) intestine extracts of *A. baerii* fingerlings fed the test diets for 40 days.

| Intestine Tract | Diet | Total Alkaline Protease | Trypsin | Chymotrypsin | Leucine Aminopeptidase | Alkaline Phosphatase |
|------|------|------|------|------|------|------|
| PI | CT | 618.8 ± 27.9 a | 1.66 ± 0.31 | 8.31 ± 0.64 | 0.16 ± 0.01 b | 15.8 ± 1.2 |
| PI | SC 10 | 713.9 ± 55.3 b | 1.90 ± 0.04 | 8.21 ± 0.63 | 0.10 ± 0.01 a | 15.1 ± 1.7 |
| DI | CT | 673.8 ± 41.9 a | 2.17 ± 0.20 | 9.85 ± 0.04 b | 0.27 ± 0.04 b | 13.35 ± 1.67 |
| DI | SC 10 | 771.5 ± 25.7 b | 2.01 ± 0.38 | 7.18 ± 0.26 a | 0.11 ± 0.02 a | 11.84 ± 1.11 |

Values are mean ± SE of triplicate determination. Values in the same column with different lowercase indicate significant difference ($p < 0.05$) ($n = 6$).

The most dominant enzyme of the intestinal brush border is the alkaline phosphatase and is commonly used as general marker of nutrient absorption [50]. In addition, the observed increase of total proteolytic activity level might contribute to compensate the lower leucine aminopeptidase activity in SC 10 group yielding similar growth performance in both groups, which confirmed adequate utilization of dietary nutrients despite the differences found in those digestive enzymes.

### 3.7. Analysis of Intestinal Mucosa

Visual analysis of SEM and TEM images confirmed that none of microalgae-fed fish showed structural damage in microvilli compared to control fish (Figures 3 and 4); the existence of detrimental changes could hinder its function as nutrient absorption surface and natural barrier against pathogenic bacteria. Intestinal mucosa showed numerous mucous cells as well as columnar ciliated and non-ciliated cells.

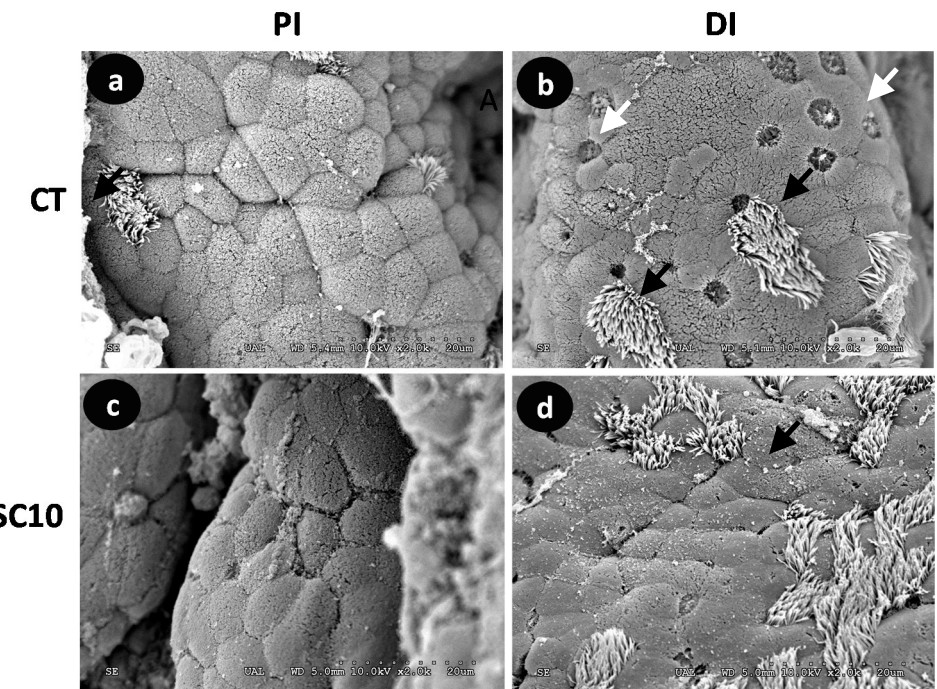

**Figure 3.** Comparative SEM micrographs from the intestine of *A. baerii* fingerlings fed the test diets for 40 days. (**a**) Proximal intestine (PI) of fish fed control (CT) diet; (**b**) Distal intestine (DI) of fish fed control (CT) diet; (**c**) Proximal intestine (PI) of fish fed microalgae-supplemented (SC 10) diet; (**d**) Distal intestine (DI) of fish fed microalgae-supplemented (SC 10) diet. In this species, ciliated cells (black arrows) can be seen alternating with goblet (white arrows) and columnar absorptive cells. SEM bar scale: 20 μm.

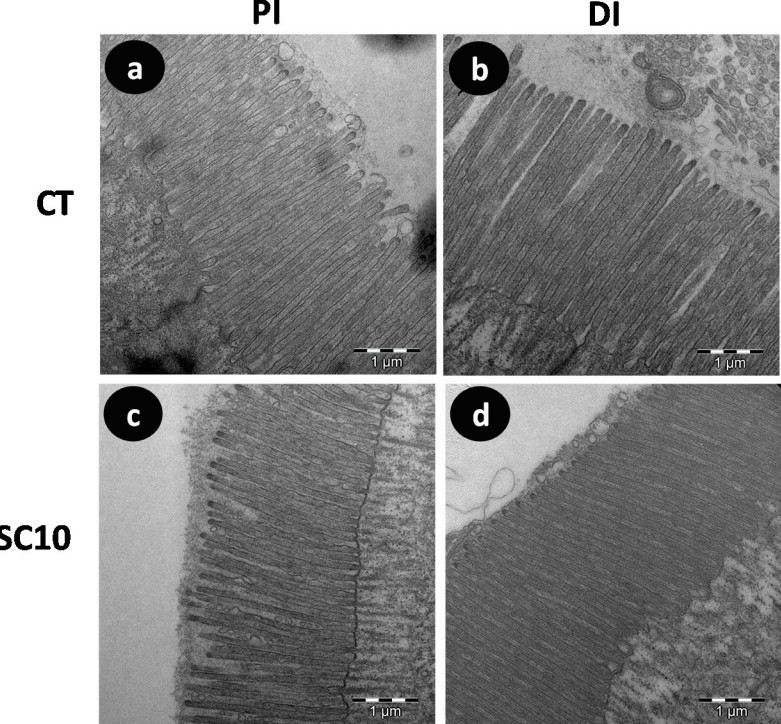

**Figure 4.** Comparative TEM micrographs detailing microvilli from the intestine of *A. baerii* fingerlings fed the test diets for 40 days. (**a**) Proximal intestine (PI) of fish fed control (CT) diet; (**b**) Distal intestine (DI) of fish fed control (CT) diet; (**c**) Proximal intestine (PI) of fish fed microalgae-supplemented (SC 10) diet; (**d**) Distal intestine (DI) of fish fed microalgae-supplemented (SC 10) diet. TEM bar scale: 1 μm.

Morphometric analysis of TEM images showed that specimens fed on microalgae-supplemented aquafeeds presented shorter and wider microvilli than fish fed the control diet, which yielded higher apical absorption surface in mucosa of distal intestine (Table 10). This analysis is a useful tool for assessing how changes in the diet may influence the structure and morphology of intestinal mucosa as it has been described that the use of vegetable proteins causes alterations in gut morphology of fish with detrimental consequences on the digestive physiology and absorption capacity [51,52]. In the present study, fish fed both experimental diets showed a healthy intestinal mucosa, which agrees with the growth performance values obtained. The results obtained showed that the dietary inclusion of microalgae from a biorefinery does not exert negative effects on intestinal mucosa since the absorption surface in enterocytes remains unaffected, and even significantly increased in the distal intestine.

**Table 10.** Microvilli morphology in proximal (PI) and distal (DI) intestine of *A. baerii* fingerlings fed the test diets for 40 days.

| Intestine Tract | Diet | Microvilli Length ($\mu$m) | Microvilli Diameter ($\mu$m) | Apical Absorption Surface per $\mu m^2$ |
|---|---|---|---|---|
| PI | CT | 2.40 ± 0.02 b | 0.10 ± 0.02 | 65.42 ± 0.67 |
| PI | SC 10 | 2.32 ± 0.02 a | 0.12 ± 0.02 | 66.09 ± 0.50 |
| DI | CT | 3.48 ± 0.03 b | 0.11 ± 0.01 | 65.87 ± 0.63 a |
| DI | SC 10 | 2.97 ± 0.02 a | 0.12 ± 0.02 | 74.42 ± 0.52 b |

Values are mean ± SE of triplicate determination. Values in the same column with different lowercase indicate significant difference ($p < 0.05$) ($n = 6$).

## 4. Conclusions

Few studies have been published on microalgae-derived feed ingredients in Acipenserids diet and no data are available on the use of biomass from biorefineries in aquafeed for this species. This is the first study on this topic. The data observed confirm the potential use of the microalgae blend *Scenedesmus-Chroococcus,* cultivated on digestate, for feeding Siberian sturgeon fingerlings. These results suggest that microalgal biomass obtained from a biorefinery using digestate liquid fraction as source of nutrients, could be a valuable alternative ingredient for ensuring a diet fulfilling the nutrient requirements of Siberian sturgeon, ensuring an adequate growth performance and fillet quality. Experimental diet supplemented with these microalgae fulfils the nutrient requirements for ensuring proper growth performance, adequate fillet quality and a healthy gastrointestinal tract in fish. However, the presence of some *C. perfrigens* spores in biomass suggests the need for further studies on pre-treatment of centrate, for instance, by thermal energy or membrane-based technology such as ultra filtration combined with reverse osmosis, to improve the microbiological quality of biomass resulting from the biorefinery.

Furthermore, in order to guarantee the ingredient composition of experimental diets, it would be interesting to monitor the biomass chemical composition of algae for at least one year, over the four seasons, so that it would be possible to check if some differences in the macronutrient composition of algae could occur despite the standardization of the cultivation protocol.

**Author Contributions:** Conceptualization, P.K. and E.F.; methodology, L.F., L.P., A.P.; validation, M.V., A.L., A.J.V., F.J.A., D.C., A.T.; formal analysis, M.V., A.L., A.J.V., F.J.A., D.C., A.T.; investigation, T.B.; resources, T.B.; data curation, P.K. and E.F.; writing—original draft preparation, T.B.; writing—review and editing, T.B. and K.P.; visualization, T.B.; supervision, P.K. and E.F.; project administration, P.K. and E.F.; funding acquisition, P.K. and E.F. All authors have read and agreed to the published version of the manuscript.

**Funding:** The research was funded by Fondazione Cariplo, under the project "Polo delle Microalghe" grant number 1411 (2015) and by the Italian Ministry of Education, University and Research (MIUR) under the project "VADEMECUM".3.

**Conflicts of Interest:** The authors declare no conflict of interest.

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
