# Peer review of "Microalgae from Biorefinery as Potential Protein Source for Siberian Sturgeon (A. baerii) Aquafeed"

_sustainability, doi:10.3390/su12218779_

Round 1

Reviewer 1 Report

The results of this paper are very interesting and can be easily used in aquaculture and in Integrated Multitrophic Acquaculture.

The approach proposed for the authors is innovative.

Author Response

Thank you very much!

Reviewer 2 Report

This article provides a potential approach for the use of microalgal in aquafeed. The utilization of the digestate from pig farm is a promising way to deal with the biowaste generated from agriculture. It was found that the microalgae could be a protein feedstock in Siberian sturgeon aquafeed with an environmental benefit in bioremediation removal of hazardous metals and pollutants from the digestate. However, I have a few concerns before the publication of this article.

  • How you control the ingredients composition of experimental diets, if the chemical composition of the ingredients varied, how you can guarantee the influence of the flexible compositions on the microalgal growth and the protein source?
  • How often and how much you collect the digestate from pig farm, and how you pretreat the digestate to control the quality?
  • How you select the strain of the microalgae?
  • Please add statements on future outlook and the limitations of this work
  • If any data related to QA/QC are available, please include in the text.
  • Please examine the article format to meet the journal requirement.

Reviewer 3 Report

Authors presented an extensive analysis of the use of microalgae from biorefinery for fish feed, claiming that it would reduce the disposal costs for microalgae production. Paper is the first study on this topic and gives huge amount of reliable results. Below, I propose two minor comments in order to improve the manuscript:

(1) In Subsection 3.1 (line #297):
Could you explain (more specifically) "suboptimal characteristics"?

(2) In Subsection 3.1 (line #311): There are different units for quantifying the algal biomass (10^6 cell mL-1), i.e. not in g/L. Why?
